# Physical Quality of Life of Sepsis Survivor Severely Malnourished Children after Hospital Discharge: Findings from a Retrospective Chart Analysis

**DOI:** 10.3390/life12030379

**Published:** 2022-03-05

**Authors:** Monira Sarmin, Tahmina Alam, Nusrat Jahan Shaly, Didarul Haque Jeorge, Farzana Afroze, Lubaba Shahrin, K. M. Shahunja, Tahmeed Ahmed, Abu Sadat Mohammad Sayeem Bin Shahid, Mohammod Jobayer Chisti

**Affiliations:** 1Nutrition and Clinical Services Division, International Centre for Diarrhoeal Disease Research, Dhaka 1212, Bangladesh; drmonira@icddrb.org (M.S.); drtahmina@icddrb.org (T.A.); nusrat.jahan@icddrb.org (N.J.S.); didarul.haque@icddrb.org (D.H.J.); farzanaafroz@icddrb.org (F.A.); lubabashahrin@icddrb.org (L.S.); tahmeed@icddrb.org (T.A.); sayeem@icddrb.org (A.S.M.S.B.S.); 2Institute for Social Science Research, The University of Queensland, Brisbane 4072, Australia; k.shahunja@uq.net.au

**Keywords:** follow-up, hospitalization, mortality, quality of life, sepsis

## Abstract

Background: Quality of life (QoL) among pediatric sepsis survivors in resource-limited countries is poorly understood. We aimed to evaluate the QoL among sepsis survivors, by comparing them with non-sepsis survivors three months after hospital discharge. Methodology: In this retrospective chart analysis with a case–control design, we compared children having sepsis and non-sepsis at hospital admission and during their post-hospitalization life, where the study population was derived from a hospital cohort of 405 severely malnourished children having pneumonia. Results: The median age (months, inter-quartile range) of the children having sepsis and non-sepsis was 10 (5, 17) and 9 (5, 18), respectively. Approximately half of the children among the sepsis survivors had new episodes of respiratory symptoms at home. Though death was significantly higher (15.8% vs. 2.7%, *p* ≤ 0.001) at admission among the sepsis group, deaths during post-hospitalization life (7.8% vs. 8.8%, *p* = 0.878) were comparable. A verbal autopsy revealed that before death, most of the children from the sepsis group had respiratory complaints, whereas gastrointestinal complaints were more common among the non-sepsis group. Conclusions: Pediatric sepsis is life-threatening both during hospitalization and post-discharge. The QoL after sepsis is compromised, including re-hospitalization and the development of new episodes of respiratory symptoms especially before death.

## 1. Introduction

Sepsis, resulting from the dysregulated host reaction to infection, often progresses to severe sepsis, septic shock, and organ dysfunction syndrome and is responsible for one in five deaths worldwide [1,2]. A recent study reflected that in 2017, there were 48.9 million cases of sepsis accounting for 11 million deaths worldwide. Half of sepsis is prevalent in children, causing 2.9 million deaths [3]. Over time, recognition of sepsis is increasing, with a decreasing trend of death [3,4,5]. However, this statement has attracted criticism due to the variation in nomenclatures and the diagnostic criteria of sepsis in different settings. Both developed and developing countries face an enormous burden of sepsis. Though publications are scarce, Bangladesh, being a developing country, also faces the same burden of sepsis with high mortality and morbidity [6,7]. The Dhaka Hospital of International Centre for Diarrhoeal Disease Research, Bangladesh (icddr,b) manages almost 200,000 diarrheal patients in a year. More than half of these patients are children under five years of age [8]. Diarrhea and pneumonia are the most common causes of sepsis and sepsis-related deaths [3]. Several studies from icddr,b have identified sepsis and its consequences as important causes of death. Malnutrition, pneumonia, and electrolyte disorder from diarrhea make children more vulnerable to developing sepsis or adding fuel to sepsis, where it turns to septic shock, often with fatal outcomes [9,10,11].

Sepsis can cause acute lung injury [12], critical illness weakness [13], and delirium [14] in acute care settings. The consequence of sepsis does not end with the recovery from acute illness even at post-discharge of a child. This highly lethal disease also increases the likelihood of death of a sepsis survivor over the months following discharge from the hospital [15], and negatively influences the quality of post-discharge life [16]. The mechanism is not well understood. There is prolonged immunosuppression with impairment of cytokine secretion and dysfunction of T-cells [17]. It is postulated that sepsis adversely involves different organ systems including the immune, CNS, psychiatric, cardiovascular, and renal systems [15,17]. It influences the neurocognition, physical and mental growth of a child and results in chronic disabilities and hospital readmission [18,19,20]. Thus, for the last two decades, post-sepsis syndrome has been gaining interest among clinicians and researchers.

As an essential health outcome, health-related quality of life (QoL) contains different dimensions of health, including physical, mental, and social [20,21,22]. It is based on the individual’s perception of his/her health status, wellbeing, expectations, pain, and sleep. This concept of quality of life is more subjective [21]. It is impaired following critical illness [23]. Different tools (visual analogue scale, questionnaire for QoL involving caregivers) are used to assess pediatric patients’ QoL. However, data are limited in assessing post-discharge outcomes among severely malnourished children having concomitant sepsis.

This study aims to evaluate the physical quality of life (mortality and morbidity) in sepsis survivors, comparing them with non-sepsis survivors three months after hospital discharge. The secondary objective is to evaluate the potential cause of post-discharge deaths among sepsis survivors using verbal autopsy.

## 2. Materials and Methods

### 2.1. Ethics Statement

For this study, we reviewed the archived medical records of the intended group without any direct interaction with caregivers. Deidentified data were used for analysis. Nevertheless, ethical approval was sought from the Institutional Review Board of icddr,b.

### 2.2. Study Setting

This study was conducted at the Dhaka Hospital of icddr,b. icddr,b is an international health research institute based in Bangladesh. Dhaka Hospital provides treatment for around 200,000 diarrheal patients in a year. Among them, 57% are under the age of 5 years. Patients are admitted to the ICU if they have severe sepsis, or septic shock, or impaired consciousness, or convulsion, or severe pneumonia with hypoxemia or respiratory failure along with diarrhea. We have facilities to provide vasoactive ionotropic support and mechanical ventilation if required. We continue to provide life-saving services to the people of Bangladesh free of cost [24].

### 2.3. Study Design

This secondary analysis used the data set primarily collected for a prospective study to identify the etiology of pneumonia among severely malnourished children [25]. In the main study, children of either sex with severe malnutrition were recruited, aged 0–59 months, with WHO-defined clinical as well as radiological pneumonia. All were admitted into either the Intensive Care Unit (ICU) or Acute Respiratory Infection (ARI) ward of the Dhaka Hospital of icddr,b between April 2011 and June 2012. From the initial cohort, we included children who visited icddr,b for a 12-week follow-up. We excluded children who required referral from the hospital during their index admission or left against medical advice. We distinguished children who had the feature of sepsis according to surviving sepsis guidelines and compared them with the non-sepsis group.

We found 405 children from the primary/index cohort who were hospitalized for severe malnutrition and pneumonia. During hospital course, 183 had sepsis and 222 had no sepsis. Among them, 29 and 6, respectively, died in the hospital. Another 34 children required either referral to another hospital or left against medical advice. At discharge from the hospital caregivers of 370 children were counseled to visit the nutrition follow-up unit (NFU) at a pre-scheduled interval. Among them, 56 did not visit NFU for further follow-up and 31 died at home. Finally, 283 children came for NFU, among them 121 from the sepsis group and 162 from the non-sepsis group (Figure 1). These 283 children constituted our study participants for whom we aimed to explore the physical QoL after sepsis. Physical QoL was assessed from available information at 12 weeks of follow-up.

### 2.4. Patient Management

While in-patients, we followed standard guidelines of the Dhaka Hospital of icddr,b, which were prepared following WHO guidelines and included the standard of care for diarrhea, sepsis, rehydration, nutritional rehabilitation, and management of pneumonia and hypoxemia (defined as SpO_2_ < 90% at sea level). Antibiotics were changed according to hospital protocol if clinical conditions demanded. Supportive care was provided as and when required. We need to acknowledge that diagnosis and management of sepsis are evolving, with newer investigations such as lactate being proposed for early diagnosis. However, the principles of management are almost the same as before—fluid bolus, antibiotics, inotropes, and vasopressor to restore BP and urine output. Dhaka Hospital has facilities to provide vasoactive-inotropic support, non-invasive respiratory support (self-invented Bubble Continuous Positive Airway Pressure), and mechanical ventilation if required [24]. During discharge from the hospital, severely malnourished children received micronutrients supplementation and health education as per protocol and were also scheduled for a regular visit to follow their health status.

### 2.5. Measurements

Our outcome of interest was the comparison of post-sepsis life between sepsis survivors and non-sepsis survivors at a 12-week follow-up following discharge from the hospital. Further comparison of these two groups was made on their admission clinical condition, anthropometry status, comorbidities, severity score and treatment. A modified Liverpool quick Sequential Organ Failure Assessment (LqSOFA) score for children was used to assess the severity of the study children at admission to the hospital (Appendix A).

We also aimed to explore the probable cause of death at home of those children who died before attending the 12-week follow-up visit at the hospital. We recorded all clinical and demographic data in a case record form designed, pre-tested and finalized for this study.

### 2.6. Operational Definitions

Severe Malnutrition: severe wasting [weight for length/height Z score <−3 of the median of the WHO anthropometry (WLZ/WHZ)] or the presence of nutritional edema or weight for age Z score <−4 of the median of the WHO anthropometry (WAZ) indicate severe malnutrition. Severe wasting is acute in origin. Severe underweight is a form of severe malnutrition that can be either acute or chronic [25,26,27].

Sepsis: In the background of infection, any two of the following is sepsis: age-specific tachycardia, hyperthermia (≥38.5 °C), aberration of white blood count.

Severe sepsis: Sepsis leading to poor peripheral perfusion evident by age-specific hypotension and/or absent peripheral pulses and/or delayed capillary refilling time in absence of dehydration.

Septic shock: Clinical condition when children are unresponsive to crystalloid (normal saline/cholera saline) boluses (20 mL/kg with a maximum of 40 mL/kg over 2 h), requiring inotropic support [28].

Liverpool pediatric quick Sequential Organ Failure Assessment (LqSOFA): LqSOFA was populated by using mental status, respiratory rate, heart rate, and capillary refill time of the study children at enrolment to the index study. It is a severity score (Appendix A) [29].

Verbal autopsy (VA): A structured interview with the caregivers of the deceased to find the cause of death, mostly used for deaths in communities where physician certification is not available. It helps to identify age–sex–time-standardized mortality rates and trends across countries. These data are crucial in saving lives, as VA provides unanswered information [30]. For the index study, the WHO VA questionnaire was used. A verbal autopsy questionnaire for the index study was developed following the WHO verbal autopsy questionnaire. A trained member of the research staff visited the home of the deceased child with the caregiver’s permission and interviewed them to obtain the information. They communicated with the study physician if required during the interview (Appendix A).

### 2.7. Analysis

All data were entered into SPSS for Windows (IBM SPSS Statistics for Windows, Version 20.0. Armonk, NY: IBM Corp) and Epi-Info (version 7.0, Epi Info™ software; Center for Disease Control and Prevention, Atlanta, GA, USA). Differences in proportion were compared using the Chi-square test. Student’s t-test was used to compare the means of homogenous data, and the Mann–Whitney test was used for the comparison of non-homogenous data. A probability of less than 0.05 was considered statistically significant. The odds ratio (OR) and their 95% confidence intervals (CIs) described the strength of association.

## 3. Results

During hospitalization, the median age (months) of the septic and non-septic children was 10.0 (IQR 5.0, 17.0) and 9.0 (IQR 5.0, 18.0), respectively. Both groups were severely malnourished, with an admission weight (kg) (Median SD) of 5.06 ± 1.8 and 5.10 ± 1.7 for the septic and non-septic groups, respectively.

They had significantly more complaints of fever, difficulty in breathing, head nodding, lower chest wall indrawing, crackles in the lungs, dehydration, respiratory rate, hypoxia and required changes of antibiotics and inotropes for septic shock. Septic children had a significantly higher LqSOFA, a pediatric severity assessment score, than non-septic children (Table 1).

A total of 283 malnourished children were reported at NFU. Among them, 121 were sepsis survivors and 162 were non-sepsis survivors. Males were predominant in both groups. The complaints about diarrhea, respiratory infections/cough, rehospitalization and antibiotic use, and vitals at 12-week follow-up were comparable between the groups. However, the sepsis survivors had a significantly higher WAZ than the non-sepsis survivors at 12-week follow-up.

Approximately half of the children from both groups had symptoms (respiratory, diarrhea, fever), at 12-week follow-up after discharge (Table 2).

During hospitalization, there were 15.8% (29/183) and 2.7% (6/222) deaths among the sepsis and non-sepsis groups, respectively, and this was statistically significant [OR 6.78 (95% CI 2.75–16.72), *p* < 0.001]. However, during post-hospitalization, at 12-week follow-up, there was 7.8% (12/154) and 8.8% (19/216) death among the sepsis and non-sepsis survivors, respectively [OR 0.88 (95% CI 0.41–1.86), *p* = 0.878] (Figure 2).

We evaluated the children who died after discharge from the hospital by doing a verbal autopsy where the caregivers of the deceased were asked several questions. The participation was voluntary, and we recorded a response rate of 77% (24 out of 31 deaths). The median time gap between discharge and death was 28 days (IQR: 10, 37.5) and 17 days (IQR: 4, 24), respectively, for the sepsis and non-sepsis groups. During post-hospitalization life, a verbal autopsy revealed most of the children from the sepsis survivor group had respiratory complaints before fatality, whereas gastrointestinal complaints were more common among the non-sepsis survivor group. Most of the children did not receive any treatment before their death, and all lost weight. Among the deceased, females were predominant (Figure 3).

## 4. Discussion

We followed under-five malnourished children who were included in an “etiology pneumonia” study for three months after their discharge from Dhaka Hospital of icddr,b. This included both sepsis survivors and non-sepsis survivors. During post-discharge follow-ups, we found no difference in mortality between sepsis and non-sepsis groups. A verbal autopsy revealed most of the children had a new episode of respiratory illness (sepsis group) and gastrointestinal illness (non-sepsis) before their fatality. Thus, we were able to evaluate the physical domain of QoL only.

We found 43% of children among the sepsis survivors had a new episode of symptoms (Table 2); therefore, we can assume that their quality of life was compromised. Similar findings have been described in another study, in which the physical domain was found to remain unhealthy even up to five years [31]. Septic patients might have lower muscle strength during acute illness that they may potentially fail to regain during post-sepsis life [17,32]. Similarly, 44% of non-sepsis survivors also had symptoms following their discharge. Studies found that septic children are at risk of mortality both during and after hospitalization [5]. The post-discharge mortality rate varies between 7–43% in the 1st year [15] and 82% within five years [7,33]. Nearly half of the patients admitted to the Intensive Care Unit (ICU) of the Dhaka Hospital of icddr,b present with sepsis, and despite standard care, some septic children develop septic shock [11]. The death rate was found to be as high as 40% and 69% in children with severe sepsis and septic shock, respectively, with co-morbidities such as severe malnutrition [7]. In this study, the deaths were significantly higher among the sepsis group than the non-sepsis group. Their LqSOFA score also indicates that they were in worse condition and vulnerable to death. However, after discharge, when the children were followed up at three months, we found that deaths were more frequent in the non-sepsis group than the sepsis group, although the difference was not statistically significant. The lack of statistically significant difference may be due to a shorter follow-up or the lack of a healthy comparison group. Sepsis survivors remained immunosuppressed, with a defect in immunity making them susceptible to recurrent sepsis and rehospitalization [5,34].

We found that around 19% of sepsis survivors required one or more hospitalizations within 3 months of hospital discharge. A study found that sepsis survivors developed more episodes of recurrent sepsis compared to healthy control (35% versus 4%) [35], and recurrent sepsis results in death among one-third of the sepsis survivors [35]. However, we were unable to evaluate the deceased child for recurrent sepsis as the caregivers didn`t communicate with the study physicians and died in the community.

We evaluated the anthropometry/growth trajectory of the groups at admission and follow-up and found that there was no significant difference except WAZ. Both groups were severely underweight; however, the non-sepsis group was significantly more underweight than the sepsis group.

This finding contradicts the usual findings of post-sepsis hospitalization, where most of the studies reported adverse effects of sepsis on post-hospitalization life. In comparison with other ICU patients, a few studies have found that sepsis patients do not differ in health-related quality of life [36].

For our studied group, we can explain this by the arrangements of follow-up at nutritional follow-up unit (NFU) for severely malnourished child. The malnourished child received a follow-up card at discharge from icddr,b Dhaka Hospital and attended the NFU at regular intervals, and the child received care following a structured follow-up program by a dedicated and experienced team consisting of physicians, nurses, and health volunteers. The caregivers received micronutrient supplementation, medication for deworming, and health advice for the mental wellbeing of the child. Mothers/caregivers were taught to prepare low-cost energy-dense food at home to ensure appropriate weight gain. Thus, both the sepsis and non-sepsis survivors remained under supervision, and any ailment was detected early and managed at a time that might influence the quality of after-sepsis life and help them do even better in some aspects.

We evaluated the cause of death by performing a verbal autopsy, where the caregivers were asked several questions during the interview. The participation was voluntary, and we recorded a response rate of 77%. During post-hospitalization life, a verbal autopsy revealed most of the children from sepsis group had new episodes of respiratory complaints before the fatality, whereas gastrointestinal complaints were more common among the non-sepsis group. A recent global burden of diseases study echoed the same observation. They found diarrheal illness and lower respiratory infection caused 5.9 and 3.3 million cases of sepsis in 2017 and resulted in 0.4 and 0.6 million sepsis-related deaths, respectively [3]. However, as most of the children died in the community and they did not attend any health facilities, it is difficult to ascertain the specific cause from VA.

## 5. Limitations

We admit several limitations of our study. First, as it was a secondary data analysis, all the potential variables of interest were not available. Thus, we were only able to study the physical domain of QoL among sepsis survivors compared to non-sepsis survivors. Second, the generalizability of our study findings is limited to severely malnourished children having pneumonia. Third, children were followed up for a shorter period, and not all caregivers (13.6% and 16% among the sepsis survivor and non-sepsis survivor groups, respectively) attended the scheduled follow-up, which might limit the depiction of the entire scenario of post-sepsis life including weight changes. Fourth, control children were not healthy, they had comorbidities of pneumonia and severe malnutrition without sepsis that made them vulnerable to any type of infections during their post-hospitalization life.

## 6. Conclusions

This study provides the first description of post-sepsis mortality and morbidity among severely malnourished children from Bangladesh after discharge from hospital. Pediatric sepsis is life-threatening, and also increases the future risk of death. Before death, the sepsis survivors had more respiratory complaints than the non-sepsis survivors. To reduce sepsis-related mortality and morbidity both in hospital and after discharge, we need to extend the care from hospital to community settings. A prospective study including pediatric QoL assessments questionnaire with adequate sample size is urgently required from LMIC countries including Bangladesh to investigate QoL for survivors of sepsis to provide patient-centered holistic care that may help to reduce the lost-to-follow-up and improve post sepsis QoL.

## Figures and Tables

**Figure 1 life-12-00379-f001:**
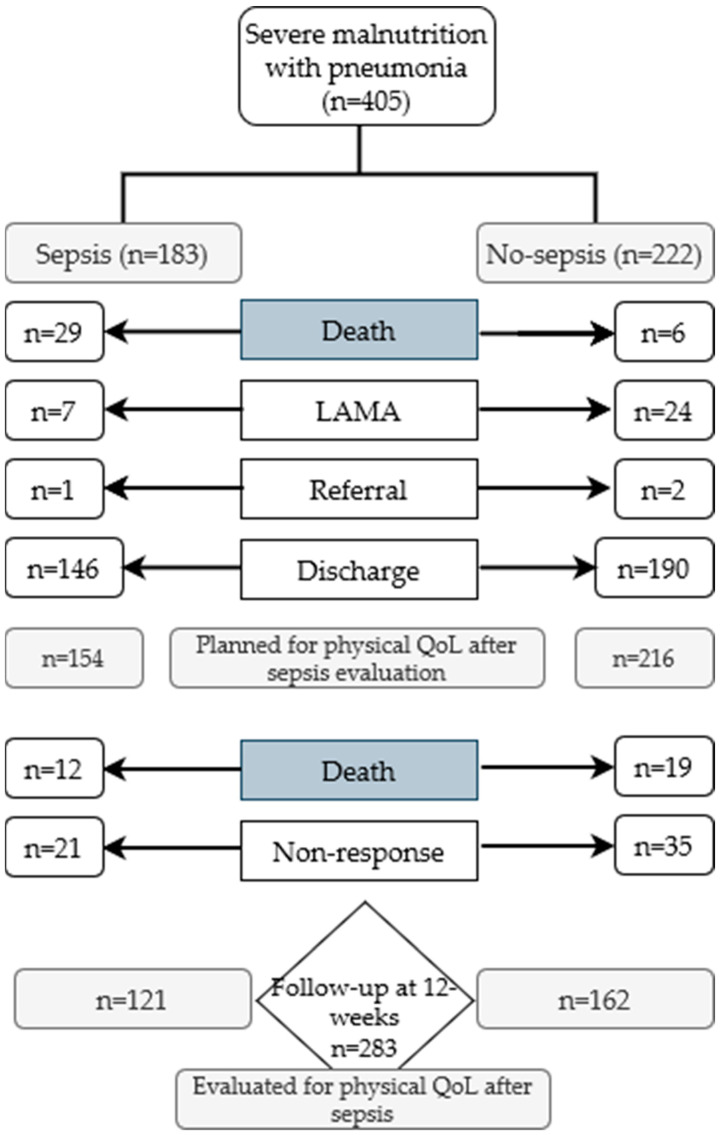
Study flow chart depicting categories of children depending on sepsis and non-sepsis. LAMA= Left Against Medical Advice.

**Figure 2 life-12-00379-f002:**
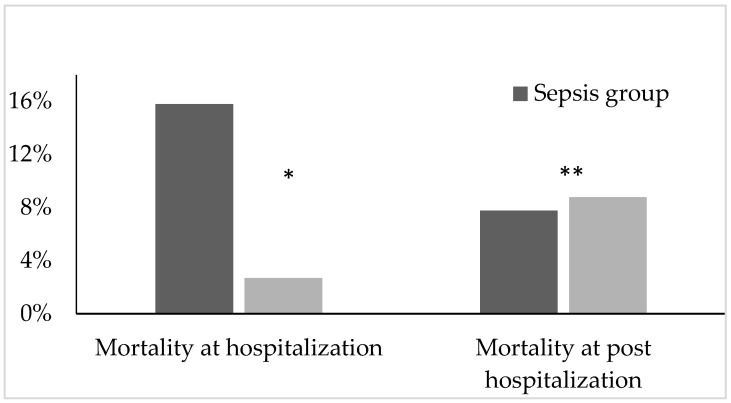
Distribution of mortality between the sepsis and non-sepsis groups both at hospitalization (* *p* < 0.001) and post-hospitalization follow-up (** *p* = 0.878).

**Figure 3 life-12-00379-f003:**
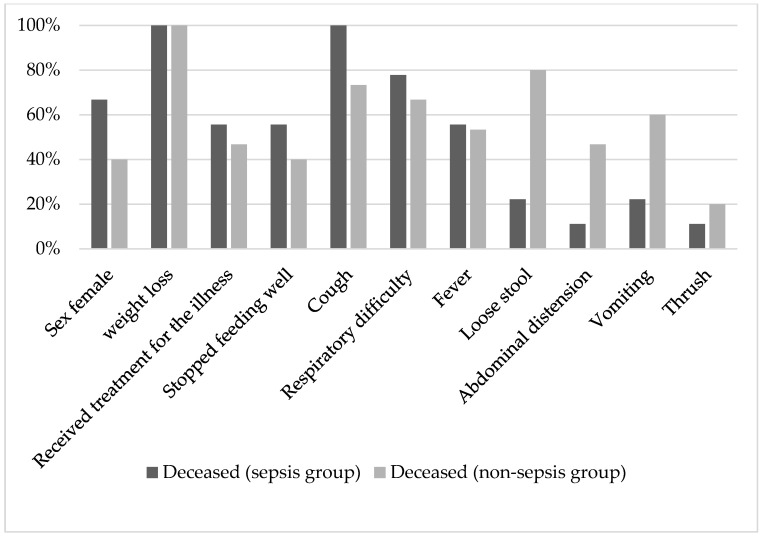
Verbal autopsy to explore the condition of the deceased child before the fatal outcome during post-hospitalization life.

**Table 1 life-12-00379-t001:** Admission characteristics of the initial cohort (children having SAM and pneumonia) grouped as having sepsis and non-sepsis.

Characteristics	Sepsis n = 183 (%)	Non-Sepsis n = 222 (%)	OR (95% CI)	*p*-Value
Sex, female	76 (41.5)	100 (45.0)	0.87 (0.58, 1.29)	0.542
Age in months (median, IQR)	10.0 (5.0, 17.0)	9.0 (5.0, 18.0)		0.884
Sociodemographic				
Poor	151(82.5)	191 (86.0)	0.77 (0.45, 1.31)	0.403
Not exclusively breast fed	118 (64.5)	165 (74.3)	0.63 (0.41–0.96)	0.041
Any illness since birth	41 (22.4)	58 (26.1)	0.82 (0.52, 1.29)	0.452
H/O prior hospitalization	28 (15.3)	32 (14.4)	1.07 (0.62–1.86)	0.913
Received antibiotic prior to hospitalization	25 (13.7)	24 (10.8)	1.27 (0.69, 2.31)	0.533
Anthropometry				
Weight (mean ±SD)	5.06 ± 1.76	5.10 ± 1.69		0.794
Height (mean ±SD)	64.31 ± 10.22	64.4 ± 10.08		0.905
WLZ (Median, IQR)	−3.61 (−4.65, −3.07)	−3.75 (−4.53, −3.05)		0.737
WAZ (Median, IQR)	−4.76 (−5.98, −3.97)	−4.94 (−5.92, −4.10)		0.658
Presenting complaints				
Fever	134 (73.2)	97 (43.7)	3.52 (2.31, 5.37)	<0.001
Cough	163 (89.1)	190 (85.6)	1.37 (0.76, 2.49)	0.371
Diarrhea	134 (73.2)	170 (76.6)	0.84 (0.53, 1.31)	0.508
Vomiting	27 (14.8)	26 (11.7)	1.30 (0.73, 2.32)	0.449
H/O poor feeding	16 (8.7)	15 (6.8)	1.32 (0.64, 2.75)	0.575
Difficulty breathing	94 (51.4)	78 (35.1)	1.95 (1.31, 2.91)	0.001
Examination findings				
Edema	12 (6.5)	10 (4.5)	1.49 (0.63, 3.53)	0.492
Dehydration	33 (18.0)	19 (8.6)	2.35 (1.29, 4.29)	0.007
Irritability	4 (2.2)	2 (0.9)	2.46 (0.45, 13.57)	0.416
Convulsion	7(3.8)	5 (2.3)	1.73 (0.54–5.53)	0.526
Lower chest wall indrawing	94 (51.4)	74 (33.3)	2.11 (1.41, 3.16)	<0.001
Nasal flaring	5 (2.7)	3 (1.4)	2.05 (0.48, 8.69)	0.476
Head nodding	6 (3.3)	1 (0.5)	7.49 (0.89, 62.80)	0.049
Respiratory rate (Median, IQR)	52 (40, 60)	40 (36, 50)		<0.001
Crackles on auscultation	133 (72.7)	125 (56.3)	2.06 (1.36, 3.14)	0.001
Wheeze				
Hypoxia (SpO_2_ < 90% in room air)	28 (15.3)	12 (5.4)	3.16 (1.56–6.41)	0.001
Hospital course				
Required changes of antibiotics	74 (40.4)	48 (21.6)	2.46 (1.59–3.80)	<0.001
Required inotrope(s)	29 (15.8)	0	-	<0.001
Hospital Acquired Infection	9 (4.9)	12 (5.4)	0.91 (0.37, 2.19)	0.996
LqSOFA ^1^ (Median, IQR)	1 (0, 2)	1 (0, 1)		0.024

^1^ Liverpool quick Sequential Organ Failure Assessment.

**Table 2 life-12-00379-t002:** Characteristics of the study children at 12-week follow-up at the nutritional follow-up unit (NFU).

Characteristics	Sepsis Survivor n = 121 (%)	Non-Sepsis Survivor n = 162 (%)	OR (95% CI)	*p* Value
Male Sex	70 (57.9)	90 (55.6)	1.09 (0.68–1.77)	0.791
Symptoms at home				
None	69 (57.0)	90 (55.6)	Reference	
Respiratory/cough	38 (31.4)	53 (32.7)	0.94 (0.56–1.58)	0.905
Fever	3 (2.5)	8 (4.9)	0.49 (0.13–1.91)	0.358
Diarrhea	5 (4.1)	7 (4.3)	0.93 (0.28–3.06)	1.000
Admission into the hospital within last 3 months				
None	98 (81.0)	137 (84.6)	Reference	
One time	18 (14.9)	16 (9.9)	1.57 (0.76–3.23)	0.292
≥Two times	5 (4.2)	9 (5.6)	0.77 (0.25–2.39)	0.783
Antibiotic use within last 3 months	22 (18.2)	29 (17.9)	1.02 (0.55–1.88)	0.924
Radial pulse rate/min (mean, SD)	132.3 ± 11.8	131.7 ± 10.2		0.666
Heart rate/min (mean, SD)	132.3 ± 11.8	131.7 ± 10.2		0.659
Respiratory rate/min (mean, SD)	40.2 ± 6.5	40.6 ± 7.2		0.645
Systolic blood pressure in mm of hg (mean, SD)	97.9 ± 10.5	97.8 ± 10.4		0.881
Diastolic blood pressure in mm of hg (mean, SD)	62.2 ± 9.4	61.0 ± 8.9		0.294
Capillary refilling time (CRT) (mean, SD)	1.9 ± 0.2	1.9 ± 0.3		0.052
Axillary temperature in Celsius scale (mean, SD)	36.9 ± 0.4	36.9 ± 0.4		0.670
SPO_2_ in room air (mean, SD)	98.6 ± 1.7	98.6 ± 2.8		0.978
Weight in kg (mean, SD)	7.0 ± 1.8	6.6 ± 1.6		0.063
Height in cm (mean, SD)	69.8 ± 8.9	67.9 ± 8.6		0.076
Weight change (kg) (mean, SD)	1.60 ± 0.9	1.53 ± 0.7		0.522
WLZ score (median, IQR)	−1.78 (−2.52, −1.18)	−1.80 (−2.73, −0.89)		0.809
WAZ score (median, IQR)	−3.14 (−4.03, −2.34)	−3.55 (−4.32, −2.62)		0.024

## Data Availability

Institutional Review Board, icddr,b has the right to share data upon request. The data request may be sent to Armana Ahmed (aahmed@icddrb.org), head, Research Administration.

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
