# Peer review of "Physical Quality of Life of Sepsis Survivor Severely Malnourished Children after Hospital Discharge: Findings from a Retrospective Chart Analysis"

_life, 2022, doi:10.3390/life12030379_

Round 1

Reviewer 1 Report

Physical Quality of Life of the Sepsis Survivor Severely Malnourished Children After Hospital Discharge: Findings From A Retrospective Chart Analysis

Journal: Life

Thank you for the invitation.

I suggest the authors’ affiliations be expanded more, not only the institution name

Keywords need to follow MeSH Browser, ideally to follow the alphabetical order.

Page 2/12, line 57: suggest to start abbreviating the quality of life to QOL

Page 2/12, line 60: space between illness and [23], “It is impaired following critical illness[23].”

Please start using the abbreviation of QoL, example: Page 2/12, line 65; page 3/12, line 110

I think the methodology and analysis need to be redone as I believe that with sepsis versus non-sepsis, the QoL will be biased towards those in non-sepsis. My suggestion is just to analyse the case group, not to part with the control group.

Causes of sepsis obtained are only respiratory and gastrointestinal illnesses which are mostly communicable diseases, are there other causes perhaps non-communicable diseases from this retrospective study?

According to Table 1, there are no definite disease diagnoses stated except the presenting complaints. I hope to see the final or working diagnoses stated in Table 1.

I believe among the markers to showcase a good recovery is biochemical markers such as albumin, haemoglobin, white cell count and etc. I think it will be excellent change that can be studied post-hospital discharge.

Need to add limitations to this study especially as it involves secondary data, instead of a prospective study.

Regards,

FH

Author Response

Rebuttal letter

Date:   Feb 1, 2022

To:   Academic Editor
       Life

From:    Dr Mohammod Jobayer Chisti

             Corresponding author,

             Manuscript ID: life-1556367

Subject:   Response on the comments of the Reviewers of ‘Life’ for manuscript ID life-1556367 entitled “Physical Quality of Life of the Sepsis Survivor Severely Malnourished Children After Hospital Discharge: Findings From A Retrospective Chart Analysis”

Dear Editor,

Thank you for providing us the opportunity to resubmit our manuscript following revision. We greatly appreciate helpful and precise comments from the respected reviewers and we have attempted to address them in full. We now submit the revised manuscript as well as this cover letter which provides a point-by-point response to the reviewers’ comments.

We sincerely hope that this manuscript is now suitable for publication in Life.

Thank you

Reviewer 1

Comments and Suggestions for Authors

General comment:

Monira Sarmin et al have conducted a restrospective analysis of results regarding the post-sepsis quality of life and mortality among post-discharge pediatric patients and highlight the importance of medical follow-up in pediatric patients’ survivors of sepsis. The paper is well-written and the data are well presented. My main concern is about the general significance and interest of this study, since the medical post-discharge follow-up of children with severe infection is in general granted in most countries.

Moreover, as elegantly described in the Introduction, the quality of life has not only physical but also mental and social dimensions. These characteristics were not studied in the on-study population. The limitations of this study should be added in the manuscript.

Response: Thank you so much for the encouraging comments. Countries like Bangladesh, post discharge follow-up is not well organized, thus many hospitals did not follow-up the children after discharge especially after the recovery from acute illness.

With the retrospective analysis with limited data, we were mainly able to study the physical QoL only after discharge from hospital for a septic illness among the severely malnourished U-5 children. However, we believe, these findings highlight the need for a prospective study with objectives related to all domain of QoL after sepsis. We added this draw back in the limitation section of the manuscript (page 11)

Specific comments:

How do the authors explain that the difference of post-discharge mortality was not significant between the sepsis and not sepsis survivors? Please explain. The general conclusion regarding the prevalence of mortality in post-sepsis survivors may be generalized to all pediatric patients with pneumonia and severe malnutrition.

Response: Thank you for the concern. The difference of in-hospital mortality was significantly higher among the sepsis group than non- sepsis group (p<0.001), however, the difference was not significant after discharge as the p value is 0.878. Now we have added the p value with the figure 2.

We also agree that the prevalence of mortality in post-sepsis survivors may only be generalized to pediatric patients with pneumonia and severe malnutrition.

In Discussion section, the authors underline that all survivor patients attended the NFU unit post discharge and were regularly followed up. This statement is in contrast with the results of this study, regarding the mortality risk combined with weight loss in these patients. Was there any problem of compliance?

Response: Thank you so much for pointing out this. During discharge, as per our hospital usual practice, we advised the caregivers to come for follow-up at Nutritional Rehabilitation Unit (NFU), however, 56 children (13.6% and 16% from sepsis survivor and non-sepsis survivor group respectively) did not attend the follow-up unit. Now we have addressed it in the limitation section.

Reviewer 2 Report

General comment:

Monira Sarmin et al have conducted a restrospective analysis of results regarding the post-sepsis quality of life and mortality among post-discharge pediatric patients and highlight the importance of medical follow-up in pediatric patients survivors of sepsis. The paper is well-written and the data are well presented. My main concern is about the general significance  and interest of this study, since the medical post-discharge follow-up of children with severe infection is in general granted in most countries.

Moreover, as elegantly described in the Introduction, the quality of life has not only physical but also mental and social dimensions. These characteristics were not studied in the on-study population. The limitations of this study should be added in the manuscript.

Specific comments:

How do the authors explain that the difference of post-discharge mortality was not significant between the sepsis and not sepsis survivors? Please explain. The general conclusion regarding the prevalence of mortality in post-sepsis survivors may be generalized to all pediatric patients with pneumonia and severe malnutrition.

In Discussion section, the authors underline that all survivor patients attended the NFU unit post discharge and were regularly followed up. This statement is in contrast with the results of this study, regarding the mortality risk combined with weight loss in these patients. Was there any problem of compliance?

Author Response

Rebuttal letter

Date:   Feb 1, 2022

To:   Academic Editor
       Life

From:    Dr Mohammod Jobayer Chisti

             Corresponding author,

             Manuscript ID: life-1556367

Subject:   Response on the comments of the Reviewers of ‘Life’ for manuscript ID life-1556367 entitled “Physical Quality of Life of the Sepsis Survivor Severely Malnourished Children After Hospital Discharge: Findings From A Retrospective Chart Analysis”

Dear Editor,

Thank you for providing us the opportunity to resubmit our manuscript following revision. We greatly appreciate helpful and precise comments from the respected reviewers and we have attempted to address them in full. We now submit the revised manuscript as well as this cover letter which provides a point-by-point response to the reviewers’ comments.

We sincerely hope that this manuscript is now suitable for publication in Life.

Thank you

Reviewer 2

Comments and Suggestions for Authors

I suggest the authors’ affiliations be expanded more, not only the institution name

Response: Thank you so much for the suggestion. We have added the detailed affiliation along with the institute name.

Keywords need to follow MeSH Browser, ideally to follow the alphabetical order.

 Response: Thank you so much, Now, we have followed MeSH Browser and also maintained an alphabetical order.

Page 2/12, line 57: suggest to start abbreviating the quality of life to QOL

 Response: Thank you, we have added the abbreviation.

Page 2/12, line 60: space between illness and [23], “It is impaired following critical illness [23].”

Response: Thank you for identifying the space issue. Now we have added space accordingly.

Please start using the abbreviation of QoL, example: Page 2/12, line 65; page 3/12, line 110

 Response: Thank you so much. We have added the abbreviation of QoL accordingly.

I think the methodology and analysis need to be redone as I believe that with sepsis versus non-sepsis, the QoL will be biased towards those in non-sepsis. My suggestion is just to analyse the case group, not to part with the control group.

 Response: Thank you. We can understand the concern of the respected reviewer. However, the objective of our study is to understand the QoL among sepsis survivors compared to non-sepsis survivors. Both the population had pneumonia and severe malnutrition. Thus, impact of sepsis in both the groups need to understand. We feel that the respected reviewer will agree with us to keep this analysis from this point of view.  

Causes of sepsis obtained are only respiratory and gastrointestinal illnesses which are mostly communicable diseases, are there other causes perhaps non-communicable diseases from this retrospective study?

 Response: We appreciate the reviewers concern. As Dhaka hospital mainly treat diarrhea, pneumonia, and malnutrition in children, we find the common etiology of sepsis is diarrhea, pneumonia, UTI. In the data set, non-communicable diseases were not recorded.

According to Table 1, there are no definite disease diagnoses stated except the presenting complaints. I hope to see the final or working diagnoses stated in Table 1.

 Response: Thank you so much for the query. Actually, our data set contains the components of different diagnoses. As all the children had pneumonia and malnutrition with and without sepsis or diarrhea, and these cover >90% under-five patient population in developing countries, we did not have documentation of other diagnoses in these children.

I believe among the markers to showcase a good recovery is biochemical markers such as albumin, haemoglobin, white cell count and etc. I think it will be excellent change that can be studied post-hospital discharge.

 Response: Thank you so much for the suggestion. We also agree with the reviewer. With clinical recovery, concomitant laboratory changes might give valuable information regarding life after sepsis. As we did not have laboratory reports, we were unable to report the results of albumin, hemoglobin and white cell count. Moreover, for the developing country set-ups these tests might not be available in all centers especially in resource poor settings.

Need to add limitations to this study especially as it involves secondary data, instead of a prospective study.

Response: Thank you so much. We have added the limitation as this is a retrospective data analysis we were unable to evaluate all the domain of quality of Health after discharge. (page 10)

Round 2

Reviewer 1 Report

All questions were satisfactorily answered

Reviewer 2 Report

The authors have adequately responded to my comments.